# International Survey of Practice for Prophylactic Systemic Antibiotic Therapy in Hip and Knee Arthroplasty

**DOI:** 10.3390/antibiotics11111669

**Published:** 2022-11-21

**Authors:** Thomas Parsons, Jonathan French, Takeshi Oshima, Francisco Figueroa, Thomas Neri, Antonio Klasan, Sven Putnis

**Affiliations:** 1Bristol Royal Infirmary, University Hospitals Bristol & Weston NHS Foundation Trust, Bristol BS2 8HU, UK; 2Musculoskeletal Research Unit, Southmead Hospital, University of Bristol Medical School, Bristol BS8 1QU, UK; 3Department of Orthopaedic Surgery, Asanogawa General Hospital, Kanazawa 920-8621, Japan; 4Clinica Alemana, Universidad del Desarrollo, Ainavillo 456, Concepción 7710171, Chile; 5Department of Orthopaedic Surgery, University Hospital Centre of Saint-Etienne, 42270 Saint-Etienne, France; 6Inter-University Laboratory of Human Movement Science, University Lyon—Jean Monnet, 42100 Saint-Etienne, France; 7AUVA UKH Steiermark, 8020 Graz, Austria; 8Faculty of medicine, Johannes Kepler University Linz, 4040 Linz, Austria; 9Avon Orthopaedic Centre, Southmead Hospital, Bristol BS10 5NB, UK

**Keywords:** arthroplasty, antibiotics, prophylaxis, infection, hip, knee

## Abstract

(1) Background: Prophylactic systemic antibiotics are acknowledged to be an important part of mitigating prosthetic joint infections. Controversy persists regarding optimal antibiotic regimes. We sought to evaluate current international antibiotics guidelines for total joint arthroplasty (TJA) of the hip and knee. (2) Methods: 42 arthroplasty societies across 6 continents were contacted and their published literature reviewed. (3) Results: 17 societies had guidelines; of which 11 recommended an antibiotic agent or antibiotic class (10—cephalosporin; 1—cloxacillin); 15 recommended antibiotic infusion within an hour of incision and 10 advised for post-operative doses (8—up to 24 h; 1—up to 36 h; 1—up to 48 h). (4) Conclusions: Prophylactic antibiotic guidelines for TJA are often absent or heterogenous in their advice.

## 1. Introduction

Prophylactic systemic antibiotic therapy (PSAT) has been used to reduce the incidence of surgical site infections (SSIs) for decades. Prior to routine use in total joint arthroplasty (TJA), the rate of prosthetic joint infection (PJI) was as high as 5% [1]. PJIs have potentially devastating sequalae in terms of mortality, morbidity and as a socioeconomic healthcare burden [2,3,4]. Proceedings from the International Consensus on Periprosthetic Joint Infection Meeting drew on the expert opinions of 400 delegates and review of over 3500 literature articles in 2013 to answer 10 questions about periprosthetic infection prevention [5]. The statement “a first- or second-generation cephalosporin (cefazolin or cefuroxime) should be administered for routine peri-operative surgical prophylaxis. Isoxazolyl penicillin is used as an appropriate alternative” received strong consensus of 89% agreement. Given this strong consensus one would anticipate this to be reflected in current national guidelines. However, current practice appears divided on the most effective antibiotic regimes to use, including the type, duration, and method of administration of antibiotics. Furthermore, rates of infection have been shown to vary amongst national joint registries and appear to be gradually increasing [6].

This study aims to investigate the practice of orthopaedic surgeons performing primary total hip arthroplasty (THA) and total knee arthroplasty (TKA) in the international community by surveying national orthopaedic, hip, knee and arthroplasty societies. The focus is on key variables: choice of antibiotic agent (which influences organisms targeted, absorption, distribution, length of action and side effect profile), timing of first dose (which influences when peak and trough tissue concentrations are achieved) and length of post-operative antibiotic course (which effects duration of antibiotic exposure and potential side effects).

## 2. Results

Forty-two societies across six continents (Africa, Asia, Australiasia, Europe, North America, South America) were reviewed (Table 1, Figure 1). Fourteen (33%) societies had published guidelines or statements for PSAT linked to their websites. The remaining twenty-eight were contacted via email to provide an opportunity to capture guidelines not publicly available. There were twelve (43%) responses from which three further guidelines were found. Twenty-five (60%) had no guidelines visible on their website and this was confirmed in nine cases with follow-up email responses. Eleven societies covered both THA and TKA, three covered THA and three covered TKA. The age of published guidance varied from 2011 to 2020. Multidisciplinary involvement in guideline creation was noted in six guidelines, three involved only surgeons, three named authors but not specialty background and the remaining five did not state contributions.

### 2.1. Antibiotic Agent

Ten (59%) guidelines suggested a first-generation cephalosporin such as cefazolin as first-line. One (5%) guideline recommended cloxacillin. The remainder (six, 35%) did not specify agents but suggested broad-spectrum intravenous antibiotics.

Five of ten guidelines that propose second-line agents suggest vancomycin in cases of known allergy with the remaining five proposing clindamycin as second-line treatment. The Australian Orthopaedic Association (AOA) suggest teicoplanin as another alternative.

The Australian, Philippine and Thai arthroplasty societies recommend vancomycin as the agent of choice when there is a “high risk for MRSA” defined as known to be infected or colonised with MRSA. The UK ‘Getting It Right First Time’ initiative (GRIFT) and the American Association of Hip and Knee Surgeons (AAHKS) advocate the use of “broad spectrum antibiotics” but do not specify agents.

### 2.2. Timing of Administration

Fifteen (88%) guidelines are explicit about the requirement for antibiotics at induction. Six guidelines state administration timings and, of these, five suggest infusion no more than 60 min and one recommends a maximum time of 45 min before incision. The Philippine Arthroplasty Society recommends PSAT administration before tourniquet inflation for TKA.

### 2.3. Duration of Cover

Of the ten guidelines that comment on length of post-operative length of PSAT, eight (80%) recommend no more than 24 h of cover. The New Zealand Orthopaedic Association state cover should be 24 to 36 h and the Japanese Orthopaedic Association recommend 48 h of cover.

## 3. Discussion

Our study showed a significant diversity of practice and in many cases a lack of guidance regarding prophylactic antibiotics. The difficulty in establishing such guidelines lies in differences in antibiotic strains over time [22], between geographic regions and centres [23] and even hip and knee arthroplasty [24]. Contribution to guidelines was also heterogenous with only six (35%) explicitly stating multidisciplinary involvement. This may be falsely low given the majority made no mention of contributions or only listed contributor names.

### 3.1. Antibiotic Choice

The majority of published guidelines suggest broad-spectrum intravenous antibiotics are used. In cases that provide more detailed guidelines (such as AAOS and SOS) there is a theme of ensuring that the most common organisms are covered: Staphylococcus aureus, Staphylococcus epidermidis, Escherichia coli and Proteus [25]. First- and second-generation cephalosporins lend themselves well to this end, having excellent Gram-positive cover as well as Gram-negative cover. Third generation cephalosporins are not recommended in any TJA guidelines likely due to their decreased activity against Gram-positive organisms.

In contrast, the SOF opts for cloxacillin due to its narrow spectrum active against the most common organisms associated with PJI and a low side effect profile [19]. The relatively short half-life of cloxacillin of 30 min to 1 h necessitates further infusion at 2 h after the first dose. This aligns with advice from the 2013 International Consensus Meeting which recommends further doses of antibiotics at double the antibiotic half-life, when blood loss is excessive (>2 L) or fluid resuscitation exceeds 2 L, whichever comes first [5].

In cases of beta-lactam allergy various strategies are proposed amongst guidelines. The SOF proposes it is safe to give cloxacillin if there is only a history of a limited rash and no itching [19]. If there is a history of an itchy rash or angioedema then cefotaxime (third-generation cephalosporin) is recommended. If there is a history of anaphylaxis then clindamycin is recommended. The majority of guidelines simply suggest a second-line agent such as clindamycin, vancomycin or teicoplanin if there is a history of allergy. It should be noted that the prevalence of penicillin allergy is much lower than recorded in patient records and that cross-reactivity of penicillins and cephalosporins is far lower than quoted by historic literature [26,27]. This suggests there is scope to increase the use of standard PSAT, reduce variation and avoid the pitfalls of second-line alternative regimes.

The increasing incidence of PJIs involving resistant organisms such as methicillin-resistant Staphylococcus aureus (MRSA) has led some units to use agents that target these such as glycopeptides vancomycin and teicoplanin [28] instead of or in addition to cephalosporin [29]. However, such strategies are associated with a higher risk of acute kidney injury (AKI) [29], further antibiotic resistance in the form of vancomycin-resistant enterococci (VRE) [30] and equivalent or higher absolute risk of PJI [31,32]. The use of flucloxacillin and gentamicin has also been associated with higher rates of AKI [33,34].

### 3.2. Timing of Administration

Timing of PSAT infusion was the most consistent variable among guidelines with 83% stating antibiotics should be given less than 60 min before incision when cephalosporins are used. A systematic review and meta-analysis including 54,552 patients found an increased risk of infection if antibiotics were administered after incision or more than 120 min before incision [35]. No difference was found between 60–120 min and 0–60 min administration. Vancomycin takes longer to reach adequate tissue concentrations and must be given as an infusion to reduce the risk of red man syndrome [36] hence guidelines frequently suggest this is started 60–120 min before incision.

### 3.3. Duration of Antibiotics

The majority of guidelines in our study recommend a maximum of 24 h of antibiotics postoperatively. This is in keeping with the 2013 International Consensus meeting on Periprosthetic Joint Infection [5]. The only exceptions to this were the New Zealand guidance which recommends 24–36 h of antibiotics postoperatively and the Japanese Orthopaedic Association which recommends 48 h of cover [8,12]. The duration of cover will partially be dictated by the half-life of the antibiotic administered, for example the serum half-life of cefazolin is 2 h compared to flucloxacillin which is 30 to 60 min.

There is little evidence that PSAT beyond a single peri-operative dose reduces the incidence of SSIs. A systematic review and meta-analysis of 4 RCTs including 4036 TJAs by Thornely et al. found no significant difference between post-operative PSAT (16 h to 6 days) and single dose PSAT at induction although they assessed the quality of evidence to be very low [37]. Similarly, a study involving 1367 total hip and knee arthroplasties done between 1991 and 1999 compared infection rates between those given a single 1 g dose of cefazolin to three 750 mg doses of cefuroxime and found no significant difference [38].

In 2017 the Centers for Disease Control and Prevention (CDC) issued a recommendation that no further doses of antibiotics should be given after closure of the surgical wound [39]. The AAOS expressed reluctance among their surgeons to adopt this and the AAHKS published a position statement in direct response to the CDC stating that it contradicted current international standards of care [21], referencing the proceedings of the 2013 International Consensus on PJI [5], with overall “limited evidence and study”. They add that a prospective randomized study has been commissioned to compare single dose and 24 h antibiotic prophylaxis. The AOA and Spanish Knee Society have similar position statements that although single dose prophylaxis should be sufficient, current evidence is underpowered and that future prospective studies should seek to clarify this [11,18]. Furthermore, the Dutch Hip Society guidelines reference that 90% of hospitals currently opt to provide 24 h of cover from the point of incision [4].

Prolonged antibiotic prophylaxis beyond 24 h has been shown to be ineffective at reducing SSIs and may increase antimicrobial resistance [40,41]. A multi-centre prospective study of 19 Australian hospitals found prolonged administration of antibiotics beyond 24 h showed no difference in rates of SSIs for primary TKA and THA [42]. Prospective surveillance of 2641 coronary artery bypass graft procedures found prolonged antibiotic exposure was a risk factor for antibiotic resistant organism infections [40].

When considering the antibiotic agent and duration, one should also consider the practicalities of ensuring correct administration. A study of adherence to peri-operative antibiotic guidelines in Canada showed only 32% of patients received their post-operative antibiotics as prescribed compared with 93% at the time of induction [43]. This may encourage surgeons to adopt regimes that do not rely on post-operative doses to provide adequate PSAT cover. Furthermore, increasing interest in daycase arthroplasty makes prolonged parenteral antibiotic administration problematic, again favouring long-acting agents that do not require further doses [44].

### 3.4. Limitations

Guidelines were only found for 40% of societies reviewed. There was a low response rate to email contact (43%) although this likely represents a true lack of guidelines as societies are less likely to reply giving negative confirmation. No guidelines were received or found for South America. Interestingly, a recent survey of TKA operative practice in Latin America had a high response rate of 83% and demonstrated a significant variation in practice compared to worldwide national registries which would suggest our finding of no guidance more likely to be genuine [45].

This study focused on available guidance and may not reflect actual practice. The diversity of worldwide guidelines presented in this study provides surgeons and healthcare providers with a large scope to adjust PSAT regimes within available guidelines. This is particularly applicable where guidance is more vague such as in the UK where the BOA recommends “each surgical unit should have a locally agreed policy including advice from microbiologists” [14]. Similarly, older guidelines such as those in Italy from 2011 are more likely to misrepresent current practice. However, the most recent published national guidance has been included and it was beyond the scope of this study to assess individual institution practice.

High-income countries were deliberately targeted due to the paucity of arthroplasty surgery and literature in low-income countries [46]. There is a risk this study may misrepresent international practice, however, countries with established arthroplasty practice are likely to lead those where practice is developing.

## 4. Materials and Methods

The Enhancing transparency in reporting the synthesis of qualitative research (ENTREQ) statement was used for study and manuscript construction [47]. Between 24 and 31 August 2022, 42 national and international orthopaedic society websites were reviewed (Table 1). The target was societies that included member surgeons who performed hip and/or knee arthroplasty. This comprised national orthopaedic, as well as national subspecialty hip and/or knee societies that included arthroplasty, and international societies specialising in hip and/or knee surgery that include arthroplasty.

Following review of websites for published guidelines those where no guidelines were present were contacted via direct email taken from the society website. They were asked to respond to a single question: ‘Do you have published guidelines on the use of prophylactic antibiotics during hip and/or knee replacement?’ With a follow up statement: ‘If yes, could you please provide them either as a link to your website or attached in a reply to this email, or if no, do you follow any other society or international guidelines?’ Where possible this question was also sent translated in the societies’ native language. A further follow-up email was sent on 14 September 2022 for those that did not initially respond. Guidelines were reviewed and data extracted by hand with reference to antibiotic agent, timing of administration, duration of antibiotics and alternative regimes. Multidisciplinary involvement in guideline construction was also noted where possible. Where guidelines were not in English they were translated by native speaking members of the authorship or translated with Google Translate (Google LLC, Mountain View, CA, USA).

## 5. Conclusions

Orthopaedic and arthroplasty societies worldwide demonstrate a lack of guidance and or a disparate range of recommendations for PSAT in the context of TJA. Overall, a cephalosporin such as cefalozin is the favoured agent of choice, administered 15 to 60 min before incision. The majority of guidelines suggest 24 h of antibiotic cover but further study is required to clarify optimum cover.

## Figures and Tables

**Figure 1 antibiotics-11-01669-f001:**
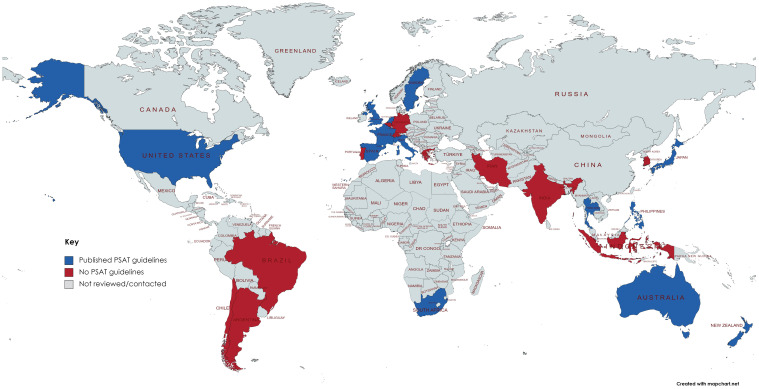
Map of countries reviewed for PSAT guidelines. Blue—published PSAT guidelines; Red—no PSAT guidelines; Grey—not reviewed or contacted. (Created using MapChart.net).

**Table 1 antibiotics-11-01669-t001:** Summary of International guidelines ordered by region in alphabetical order.

Society Name	Region	Joint (Hip/Knee)	Guideline Summary
South African Arthroplasty Society (SAAS)	Africa	Both	Intravenous antibiotics active against coagulase negative staphylococcus, according to local policy within an hour of induction, and continued for 24 h post-operatively. [7]
South African Knee Society (SAKS)		Knee	None
APKASS (Asia-Pacific Knee, Arthroscopy and Sports Medicine Society)	Asia	Knee	None
Indian Society of Hip and Knee Surgeons		Both	None
Indonesian Hip & Knee Society		Both	None
Iranian Orthopaedic Association		Both	None
Japanese Orthopaedic Association		Both	Intravenous 1st or 2nd generation cephalosporin prior to incision, continued for 48 h after surgery [8].
Korean Hip Society		Hip	None
Korean Knee Society		Knee	None
Philippine Hip and Knee Society		Both	Pre-op intravenous cefazolin 2 g, Vancomycin/Clindamycin if allergic/MRSA. Infuse before tourniquet inflation. Stop within 24 h of surgery [9].
Thai Hip and Knee Society		Both	National guidelines: intravenous cefazolin, or clindamycin/ vancomycin if allergic/MRSA. Administer within 60 min before incision. Give 2nd dose if operating time >4 h/>1.5 L blood loss. Single dose or no-longer than 24 h [10].
Arthroplasty Society of Australia (ASA) which is subspecialty group of Australian Orthopaedic Association (AOA)	Australasia	Both	Intravenous cefalozin 2 g within 60 min of incision. Vancomycin (start infusion 30–120 min before incision)/teicoplanin/clindamycin if allergic. Give 2nd dose if surgery >4 h or excessive blood loss. Single dose sufficient, 24 h maximum [11]
New Zealand Orthopaedic Association		Hip	Intravenous broad-spectrum antibiotic at induction of anaesthesia and for the first 24–36 h after the operation [12].
AGA Gesellschaft für Arthroskopie und Gelenkchirurgie (AGA-Society for Arthroscopy and Joint-Surgery)	Europe	Both	None
Arbeitsgemeinschaft für Endoprothetik (German Society for Endoprosthetics)		Both	None
Belgian Society of Orthopedics and Traumatology		Both	None
British Association for Surgery of the Knee (BASK)		Knee	Intravenous broad-spectrum antibiotics at induction of anaesthesia [13].
British Hip Society (BHS)		Hip	Intravenous antibiotics according to local policy at induction of anaesthesia [14,15].
British Orthopaedic Association (BOA)		Both	Intravenous antibiotics according to local policy at induction of anaesthesia [14,15].
ESSKA (European Society of Sports Traumatology, Knee Surgery, and Arthroscopy)		Knee	None
European Hip Society		Hip	None
European Knee Associates (EKA, subsection of ESSKA)		Knee	None
European Knee Society (EKS)		Knee	None
French Society of Hip and Knee Surgery		Both	Follow the French Society of Anaesthesia (SFAR) guidelines: intravenous cefazolin/cefuroxime or clindamycin/vancomycin if allergic. Maximum 24 h period [16].
Hellenic Association of Arthroscopy, Knee Surgery & Sports Traumatology		Knee	None
Italian Hip Society		Hip	National guidelines: 1st or 2nd generation cephalosporin, vancomycin if allergic, 30–60 min preceding incision [17].
Italian Society of Arthroscopy, Knee Surgery and Orthopaedic Sports Medicine		Knee	National guidelines: 1st or 2nd generation cephalosporin, vancomycin if allergic, 30–60 min preceding incision [17].
Portuguese Society of Orthopaedic Surgery and Traumatology		Both	None
Spanish Orthopaedic Society (Spanish Knee Society) (SOS)		Both	National guidelines: 1st or 2nd generation cephalosporin (ie, cefazolin 2 g or cefuroxime) administered intravenously within 30 to 60 min before incision as a single, weight-adjusted dose [18].
Svensk Ortopedisk Förening (Swedish Orthopedic Association; SOF)		Both	National guidelines: intravenous cloxacillin 2 g 30–45 min before incision with further doses at 2 and 6 h. Cefotaxime/clindamycin if allergic [19].
Werkgroep Heup (Dutch Hip Society)		Hip	National guidelines: intravenous cefazolin 2 g 15–60 min before incision. 2nd dose if surgery >4 h/>1.5 L blood loss. If penicillin anaphylactic: Clindamycin/vancomycin (60–120 min before incision). Not to give more than 24 h [4].
International Congress for Joint Reconstruction	International	Both	None
International Hip Society		Hip	None
ISAKOS (International Society of Arthroscopy, Knee Surgery and Orthopaedic Sports Medicine)		Knee	None
American Academy of Orthopaedic Surgeons (AAOS)	North America	Both	1st or 2nd generation cephalosporin or glycopeptide at induction taking account of institution antibiogram, patient factors and infection control expert advice. Not beyond 24 h [20].
American Arthroplasty Hip and Knee Society (AAHKS)		Both	Statement generally advocating use of postoperative prophylactic antibiotics in arthroplasty, no guidelines [21].
The Hip Society		Hip	None
Argentinian Association for the Study of the Hip and Knee	South America	Both	None
Brazilian Hip Society		Hip	None
Brazilian Society of Knee Surgery (SBCJ)		Knee	None
Latin American Society of Arthroscopy, Knee and Sports (SLARD)		Knee	None
Sociedad Chilena de Ortopedia y Traumatología		Both	None

## Data Availability

Not applicable.

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
