# Peer review of "International Survey of Practice for Prophylactic Systemic Antibiotic Therapy in Hip and Knee Arthroplasty"

_antibiotics, 2022, doi:10.3390/antibiotics11111669_

Round 1

Reviewer 1 Report

This is a survey of SSI prophylaxis practices in primary arthroplasty, collected from international orthopedic societies.

Societies provided their guidelines, which were analyzed for molecule used, timing of first dose and length of post-operative antibiotic course.

1. Results and methods sections are swapped - please change (technical editor?)

2. From the introduction is not really clear whit is the importance to "to inform on the baseline practice of orthopaedic surgeons"  "in the international community by surveying national" and "bringing together recommendations and guidelines".

Despite the heterogeneity of SSI prophylaxis, the SSI rate does not vary too much between countries with high income; it should be taken in account. However, in a middle income countries it may be different, therefore you should explain why do you decide to include all countries together.

I suggest to develop this section to bring readership to a clear understanding, why authors carried out this survey.

For instance, why the choice of molecule is important, why the timing of administration is important, and what are the reflexions behind the duration of prophylaxis.

3. Material and methods section is lean.

Please develop more, how all necessary data for the survey analysis were extracted, sorted and normalized. Was the date of publication of guidelines/latest revision date taken in account?

Please refer to the ENTREQ reporting guidelines to improve the whole manuscript flow.

Tong, A., Flemming, K., McInnes, E. et al. Enhancing transparency in reporting the synthesis of qualitative research: ENTREQ. BMC Med Res Methodol 12, 181 (2012). https://doi.org/10.1186/1471-2288-12-181

4. Results

It would be useful to plot the results from the table 1  on the World map, and also to provide the current numbers for SSI per country (if reported -should be easy to find)

It would be useful to analyze and to report the role of multidisciplinary participation into elaboration of such guidelines - usually it is acknowledged - eg surgeon, anesthesiologists, infection disease specialists, hospital hygiene or epidemiologists, microbiologists etc

5. Discussion

Well developed.

For the timing part, please refer to

de Jonge SW at al. Timing of preoperative antibiotic prophylaxis in 54,552 patients and the risk of surgical site infection: A systematic review and meta-analysis. Medicine (Baltimore). 2017 Jul;96(29)

doi.org/10.1097/md.0000000000006903

In the ATB allergy part please comment on the low prevalence of penicillin-cefazoline cross allergy (it is a very important particularity), and either this particularity is reflected in national standards

please, refer to:

Savic et al; Management of a surgical patient with a label of penicillin allergy: narrative review and consensus recommendations. British Journal of Anaesthesia, 123 (1): e82ee94 (2019) doi.org/10.1016/j.bja.2019.01.026

Vorobeichik et al ; Misconceptions Surrounding Penicillin Allergy: Implications for Anesthesiologists. Anesthesia & Analgesia: September 2018 - Volume 127 - Issue 3 - p 642-649 doi.org/10.1213/ane.0000000000003419

For the duration please comment on challenges linked with the outpatient track for TKA/UKA/THA - eg dealing with the second or third ATB injection

6. Conclusion section is weak.

Please give three statements regarding your points – molecule, timing and duration, and your opinion, how this should/should not be changed.

Author Response

Thank you for your detailed review and suggestions.

Please see the attached document detailing the changes made.

Reviewer 2 Report

I am honored to have the opportunity to review your very interesting paper.

First of all, I think Materials & Methods should be placed between Background and Results.

The research methodology is very original and informative, but I feel that the significance of this study is a bit diminished by the existence of an international consensus conference. Please describe why you decided to conduct this research despite the existence of the international consensus conference.

Is the low response rate of 40% due to the fact that those that responded that there are no guidelines were categorized as not responding? I think we can classify those who responded that there are no guidelines as respondents because they also responded.

It may be true that there is no evedence showing that prolonged use of antimicrobials reduces SSIs, but is there any data or reports showing that it promotes the creation of resistant bacteria and has some negative effects? If so, please describe it in your paper. If not, please state that there is no such data.

Perioperative antimicrobial therapy may be effective in preventing PJI that occurs immediately after surgery, but are there any guidelines regarding prevention of PJI that occurs several years after surgery? Please let me know if this is possible.

I think the paper would be more meaningful if you could describe the above.

Author Response

Thank you for your review and insightful comments.

Please see the attached document detailing the changes made.

Round 2

Reviewer 1 Report

the manuscript reads better now, I have no further remarks.